# Portuguese observational cross-sectional clinical imaging study protocol to investigate central dopaminergic mechanisms of successful weight loss through bariatric surgery

Marta Lapo Pais,[1,2,3] Joana Crisóstomo,[1,4] Antero Abrunhosa,[1,2]
Miguel Castelo-Branco  [1,2,5]

[1]Coimbra Institute for Biomedical Imaging and Translational Research (CIBIT), University of Coimbra, Coimbra, Portugal
[2]Institute for Nuclear Sciences Applied to Health (ICNAS), University of Coimbra, Coimbra, Portugal
[3]Faculty of Science and Technology, University of Coimbra, Coimbra, Portugal
[4]Champalimaud Research, Champalimaud Foundation, Lisbon, Portugal
[5]Faculty of Medicine, University of Coimbra, Coimbra, Portugal

**Correspondence to**
Dr Miguel Castelo-Branco;
mcbranco@fmed.uc.pt

## ABSTRACT

**Introduction** Bariatric surgery (BS) is the treatment of choice for refractory obesity. Although weight loss (WL) reduces the prevalence of obesity-related comorbidities, not all patients maintain it. It has been suggested that central mechanisms involving dopamine receptors may play a role in successful WL. This protocol describes an observational cross-sectional study to test if the binding of central dopamine receptors is similar in individuals who responded successfully to BS and age- and gender-matched normal-weight healthy individuals (controls). As secondary goals, the protocol will investigate if this binding correlates with key parameters such as age, hormonal status, anthropometric metrics and neurobehavioural scores. Finally, as exploratory goals, we will include a cohort of individuals with obesity before and after BS to explore whether obesity and type of BS (sleeve gastrectomy and Roux-en-Y gastric bypass) yield distinct binding values and track central dopaminergic changes resulting from BS.

**Methods and analysis** To address the major research question of this observational study, positron emission tomography (PET) with [$^{11}$C]raclopride will be used to map brain dopamine type 2 and 3 receptors (D2/3R) non-displaceable binding potential (BP$_{ND}$) of individuals who have successfully responded to BS. Mean regional D2/3R BP$_{ND}$ values will be compared with control individuals by two one-sided test approaches. The sample size (23 per group) was estimated to demonstrate the equivalence between two independent group means. In addition, these binding values will be correlated with key parameters to address secondary goals. Finally, for exploratory analysis, these values will be compared within the same individuals (before and after BS) and between individuals with obesity and controls and types of BS.

**Ethics and dissemination** The project and informed consent received ethical approval from the Faculty of Medicine and the Coimbra University Hospital ethics committees. Results will be disseminated in international peer-reviewed journals and conferences.

## STRENGTHS AND LIMITATIONS OF THIS STUDY

⇒ Investigating brain dopamine receptors in successful weight loss (WL) through bariatric surgery (BS) will ultimately help clarify how these receptors function in the context of maintenance of WL, decreased food intake and reduced motivational drive to eat after BS;

⇒ Including an exploratory longitudinal analysis employing functional positron emission tomography (PET) at two different time points, the study will be able to track BS-associated central dopaminergic neurotransmission changes;

⇒ Using functionally defined brain regions will increase the statistical power of the study analysis by defining a mean activation cluster related to visual food cue processing;

⇒ This study was designed to include only female participants, which may limit the generalisation of our results;

⇒ Exploratory subgroup analysis is limited due to the dataset size, and some level of participant dropout or exclusion is expected in the longitudinal part of this analysis.

## INTRODUCTION
### Background

Obesity is a multifactorial condition thought to result from gene–environment interactions and is mediated by complex neuronal and hormonal mechanisms.[1 2] As a result, an imbalance of energy where energy intake exceeds energy expenditure leads to weight gain.[1 2] This condition has become an epidemic of global proportions, being one of the leading causes of morbidity and mortality for the current and likely future generations.[3]

Bariatric surgery (BS) is the treatment of choice when other options have failed to treat obesity. Although weight loss (WL) reduces the prevalence of obesity-related comorbidities and improves the quality of life, not all patients manage to lose weight, and others regain it in the long term.[4–6] It has been suggested that WL after BS is likely due

to changes in central nervous hunger and satiety control rather than purely restrictive gastric volume or gastrointestinal malabsorption.[7] This implies the existence of neural mechanisms underlying BS and the role of central neurotransmission in hunger and satiety control.[8–11] Dopamine is an important neurotransmitter with implications in addiction as well as WL, decreased food intake and a reduced motivational drive to eat.[12–15] Concerning cerebral control of food intake, the dopamine type 2 receptors (D2R) are particularly interesting. Original studies on this topic have been reporting that drugs that block D2R increase appetite and result in weight gain, such as antipsychotic drugs[16] and treatment with dopamine receptor agonists with a high affinity for D2R causes WL.[17] Moreover, a study on diet manipulation found that dietary fat restriction significantly decreased D2R availability in striatal clusters of individuals with obesity.[18] Yet, the role of brain dopamine receptors in eating behaviour that consequently impacts body weight, particularly their function in the context of successful WL through BS, is neither consensual nor well addressed. For this reason, we propose to further investigate these receptors in individuals who have successfully responded to BS.

## Study aims

This protocol describes an observational cross-section study whose main goal is to investigate how brain dopamine type 2 and 3 receptors (D2/3R) function in the context of successful WL through BS. Namely, we aim to understand if the binding of these receptors is the same in individuals who responded successfully to BS and controls, age- and gender-matched normal-weight healthy individuals. To achieve that, positron emission tomography (PET) with [$^{11}$C]raclopride will be used to map the binding of these receptors in 23 individuals of both groups. Mean regional D2/3R non-displaceable binding potential ($BP_{ND}$) values will be extracted from these images, with a focus on three striatal brain regions (caudate, putamen and ventral striatum) and a mean activation cluster across individuals related to visual food cue processing identified by functional magnetic resonance imaging (fMRI).

As secondary goals, we aim to evaluate if these binding values correlate with anthropometric metrics, including waist/hip circumference, Body Mass Index (BMI), excess weight loss (EWL%), total body weight loss (TWL%) and weight regain, and key variables such as neurobehavioural scores, age and hormonal status.

We will also perform an exploratory analysis using the same molecular imaging dopaminergic system. We will compare mean regional D2/3R $BP_{ND}$ values of independent groups of individuals (sleeve gastrectomy and Roux-en-Y gastric bypass, controls and individuals with obesity) to investigate whether the type of BS and obesity display distinct dopamine receptors binding values, respectively. Finally, we will track central dopaminergic changes resulting from BS by comparing the same individuals before and after BS.

We expect this work will help clarify how central mechanisms function in the context of maintenance of WL, decreased food intake and reduced motivational drive to eat after BS. Simultaneously exploring these central mechanisms in obese phenotypes and investigating the differences between types of BS, we can contribute to understanding if the binding of these receptors depends on BMI or the type of surgery used.

## METHODS AND ANALYSIS
### Study design

This study can be split into main and secondary designs. The main design will be used to answer the major and secondary questions, whereas the secondary design will be used for exploratory research questions (see figure 1). Both studies will be conducted at the Institute for Nuclear Sciences Applied to Health (ICNAS) University of Coimbra, Coimbra, Portugal. The expected overall study duration is approximately 3 years, from January 2022 (when we obtained both approvals from ethics committees) to January 2025. We anticipate a 3-year timeframe to disseminate the study in the community and social media, collect contacts of possible participants, optimise and test the study protocol by pilot acquisitions using healthy volunteers from our volunteer database, recruit participants, acquire and analyse the data and submit the results for publication in peer-reviewed journals.

The main study design is an observational cross-sectional clinical imaging study to evaluate if the binding of D2/3R is the same in individuals who responded successfully to BS and controls. Two groups of female subjects will be included: women who have successfully responded to BS (n=23) recruited from Coimbra Hospital and University Centre (CHUC) or the Portuguese association of patients who underwent BS—*Associação Portuguesa dos Bariátricos*; and age- and gender-matched normal-weight (BMI: 18.5–24.9 kg/m$^2$) healthy individuals (n=23) recruited from our volunteer database and public advertisements. To be successful BS responders, individuals should achieve at least 50% EWL% or 20% TWL% at 1-year postop.[19] Since we will include participants from 1 to 5 years postop, we will use a retrospective approach to verify this criterion and consider the weight participants had 1-year postop. We will also guarantee that at the time of data acquisition, participants still fulfil the criterion defined for 1 year after surgery (>50% EWL% or 20% TWL%). Follow-up time postop will be further used as a covariate in the statistical analysis.

The secondary study design includes a longitudinal and a cross-sectional part. The longitudinal part was designed to track central dopaminergic changes resulting from BS. For that purpose, we will include a cohort of individuals with obesity at two different time points, before and after BS. On the other hand, in the cross-sectional part, we will split the individuals who underwent BS according to the type of surgery to investigate whether the type of BS displays distinct dopamine receptor binding values.

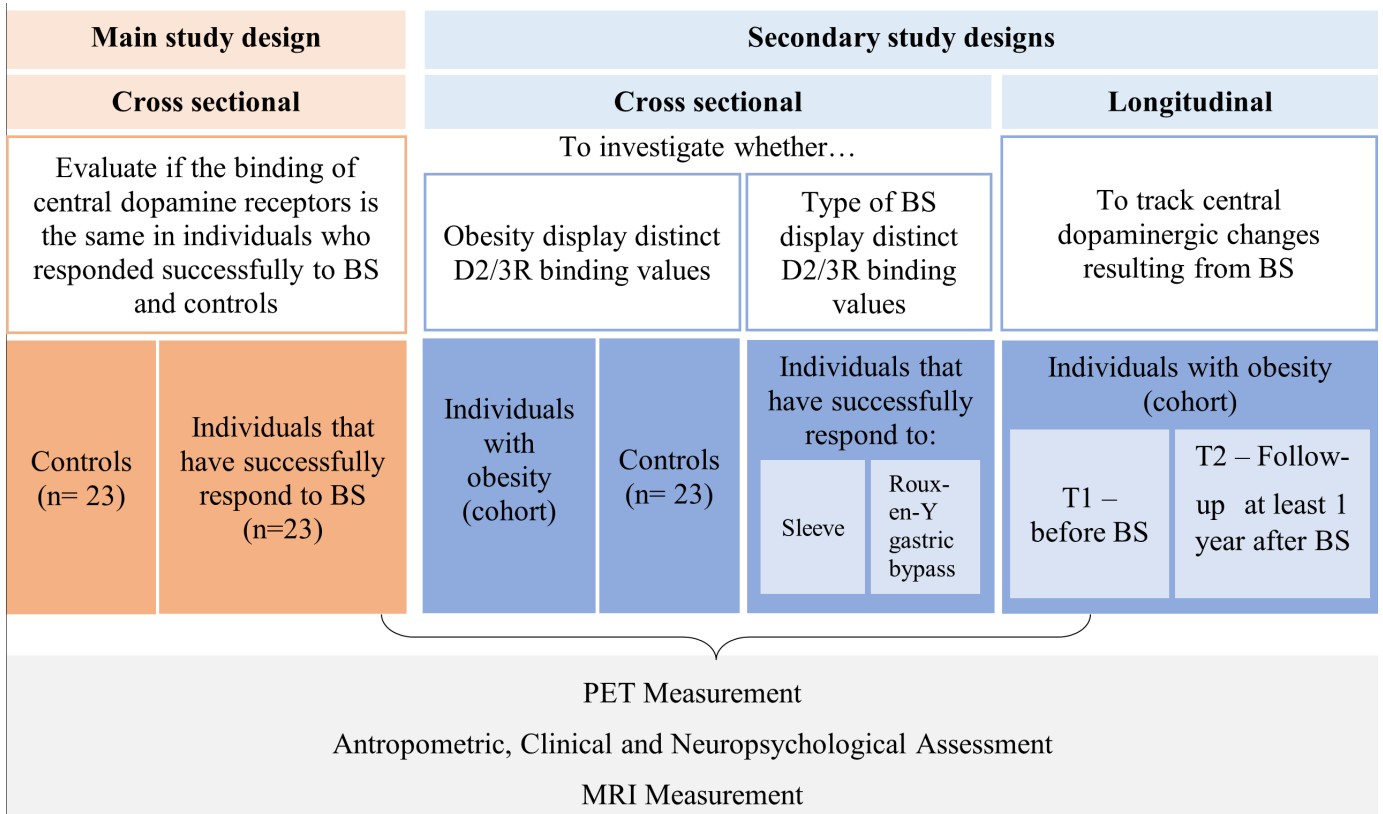

**Figure 1** Main and secondary study designs, research questions and samples. Controls are the age- and gender-matched normal-weight healthy individuals. BS, bariatric surgery; D2/3R, dopamine type 2 and 3 receptors; MRI, magnetic resonance imaging; PET, positron emission tomography.

Finally, to investigate whether these binding values are altered in obesity, we will compare the individuals with obesity of the cohort with the controls. The study designs, research questions, and samples are schematised in figure 1.

Study-specific inclusion criteria are signed informed consent, women aged between 18 and 65 years, BMI >33 kg/m² for individuals with obesity and 18.5–24.9 kg/m² for age- and gender-matched normal-weight healthy individuals. No restriction regarding BMI range will be imposed on the group who have successfully responded to BS. Exclusion criteria are previous gastrointestinal surgery, current treatment with medication interacting with central dopamine, active neoplastic or inflammatory disease, history of neurological disease, traumatic brain injury or psychiatric disorder, previous/current alcohol or other substance abuse and conditions that preclude magnetic resonance imaging (MRI) or PET.

### Patient and public involvement
Patients were not involved in the development of the research question, outcome measures or study design but were involved in the dissemination.

### PET measurement (acquisition and analysis)
Dynamic PET scan of the entire brain in 3D mode for 60 min (24 frames: 4×15 s, 4×30 s, 3×60 s, 2×120 s, 5×240 s, 6×300 s) together with computed tomography (CT) after

intravenous bolus injection of a maximum of 15 mCi of [$^{11}$C]raclopride will be acquired using PET/CT scanner (Siemens Biograph Vision 600). A dedicated and experienced team will perform [$^{11}$C]raclopride synthesis in the facilities of the ICNAS according to the methods described in Rodrigues et al.[20]

After acquisition, accurate coregistration (between PET images and their anatomical MRI images) will be performed and further confirmed by visual inspection across all planes. As binding of [$^{11}$C]raclopride is negligible in the cerebellum, simplified reference tissue models will be used to analyse PET images with [$^{11}$C]raclopride.[21] As a result, we will obtain the parametric maps of D2/3R BP$_{ND}$. These maps will be generated using the cerebellum as a reference region, and two distinct reference tissue methods will be evaluated that are Logan Plot and MRTM2 (multilinear reference tissue model 2), as described and evaluated by the methods thoroughly reported in Logan[22] and Ichise et al,[23] respectively. Mean regional D2/3R BP$_{ND}$ values of anatomical brain regions of interest (ROIs) will be extracted from the parametric maps. Due to the low signal-to-noise ratio of [$^{11}$C]raclopride in extrastriatal brain regions,[24] anatomical ROIs will be defined in three subregions of the striatum, caudate, putamen and ventral striatum. Additionally, to increase the statistical power of the study analysis, we will test a mean activation cluster resulting from fMRI analysis,

which will allow us to take into account other regions where dopaminergic receptors play an important biological role.[25] All PET data analysis will be performed using the Pmod software (PMOD, V.4.105; PMOD Technologies, Zurich, Switzerland) and SPM12 (V.12, Wellcome Trust Centre for Neuroimaging, London, UK).

Despite the lack of studies, we expect the same mean regional D2/3R $BP_{ND}$ values in individuals who have successfully responded to BS and healthy controls. We hypothesise that D2/3R $BP_{ND}$ depend on the motivation drive to eat, which we speculate to be the same in these two groups. Following the same line of thought, in exploratory analysis, we expect that obesity will be associated with altered mean regional D2/3R $BP_{ND}$ values in the brain but no differences between types of BS (sleeve gastrectomy and Roux-en-Y gastric bypass). Moreover, we expect that these dysfunctional values can be reversed to control condition values after WL. It is important to note that because [$^{11}$C]raclopride is susceptible to endogenous dopamine, D2/3R $BP_{ND}$ values do not reflect only receptor availability but also the levels of dopamine tone. A recent preprint study compared [$^{11}$C]raclopride to [$^{18}$F]fallypride, a radiotracer also susceptible to endogenous dopamine but with a higher affinity for D2/3R, and concluded that D2/3R $BP_{ND}$ values measured by PET with [$^{11}$C]raclopride are more sensitive for revealing between-group differences resulting from dopamine tone.[26]

### Clinical and neuropsychological assessment

The key variables of the clinical assessment include age, five anthropometric metrics and hormonal status. Anthropometric metrics are BMI, waist/hip circumference ratio for all groups and EWL%, TWL% and weight regain for the group of individuals who underwent BS. As all our participants are female, and since dopaminergic neurotransmission might fluctuate across age[27] and the menstrual cycle,[28] this information will be further used as a covariate in the statistical analysis. A brief neuropsychological evaluation will also be performed using the Eating Disorder Examination Questionnaire (EDE-Q) of 28 items,[29] the Three Eating Factor Questionnaire (TEFQ) of 21 items,[30] the Self-Compassion Scale (SCS) of 12 items[31] and the Depression Anxiety and Stress Scale (DASS) of 21 items, all translated and validated for the Portuguese population. EDE-Q has four subscales (Restraint, Eating Concern, Shape Concern and Weight Concern) and an overall global score. TEFQ includes three (Cognitive Restraint, Uncontrolled Eating and Emotional Eating). SCS is split into six subscales (Self-kindness, Self-judgement, Common Humanity, Isolation, Mindfulness and Overidentification) and an overall global score. Finally, DASS measures the emotional states of depression, anxiety and stress. To limit the number of parameters under investigation, for EDE-Q and SCS, we will only include the overall global score.

Authors have suggested that emotional eating or behaviour of discounting monetary rewards might provide a more accurate reflection of changes in central D2R function compared with BMI.[32 33] Following this line of thought and our hypothesis—D2/3R $BP_{ND}$ depend on the motivation drive to eat—we expect that neuropsychological scores correlate more strongly with central dopamine receptor function than anthropometric metrics. Namely, we expect to obtain a negative correlation between measures of dysfunctional eating behaviour/concern and self-compassion, depression, anxiety, stress and regional D2/3R $BP_{ND}$ values and no correlation with anthropometric metrics.

### MRI measurement (acquisition and analysis)

Structural and fMRI data will be acquired on a 3T imaging system (MAGNETOM Prisma, Siemens Medical Solutions) using a 64-channel head coil. First, we will acquire the structural MRI data using 3D-T1 weighted sequence (192 slices; echo time (TE): 3.5 ms; repetition time (TR): 2530 ms; voxel size: 1×1×1 mm; flip angle (FA): 7°; field of view (FOV): 256×256 mm). After that, we will start the block-design fMRI paradigm (6 runs of ~2 min each) using 2D-T2* weighted sequences (72 interleaved slices; TE: 37 ms; TR: 1000 ms; voxel size: 2×2×2 mm; FA: 68°; FOV: 200×200 mm). Each run comprises three visual blocks and three question blocks. In each visual block (20 s), participants will be presented with five images of low- or high-calorie foods or non-food objects from the Cross-Cultural Food Images Database.[34] For all participants, the sequence of the fMRI task is the same, where the first block switches between high- and low-calorie foods, while the block of non-food objects is interleaved in the middle. Before each visual block takes place a fixation cross (10 s) and after a question block (10 s), with a question about the visual block that took place before. Food visual blocks will be followed by a decision-making question of whether the person would eat such foods, considering both the pleasure and health implications. To answer the question, the participant will use a response button to choose one of four options: "1—I certainly wouldn't eat it", "2—I probably wouldn't eat", "3—I would probably eat" and "4—I would certainly eat it". On the other hand, non-food visual blocks will be followed by an attention question of whether the person recognises such objects. Here, the options are the following: "1—I recognise all objects", "2—I recognise almost all objects", "3—I hardly recognise any object" and "4—I don't recognise any objects". See figure 2 for details. We plan to use the decision-making questions to evaluate how people rate rewards in terms of pleasure and health concerns, and in particular to achieve integrated information on psychological components of reward, particularly liking (hedonic impact) and wanting (incentive salience). This information will be combined with the neuropsychological scores to conduct a correlation analysis with central dopamine receptor function. Finally, after the fMRI acquisition, the participants will also perform rating tasks using a horizontal, 100 mm, bidirectional scale between the extremes "I don't like it at all" and "I like it a lot" to rate all the food images from fMRI visual task in terms of preferences regardless

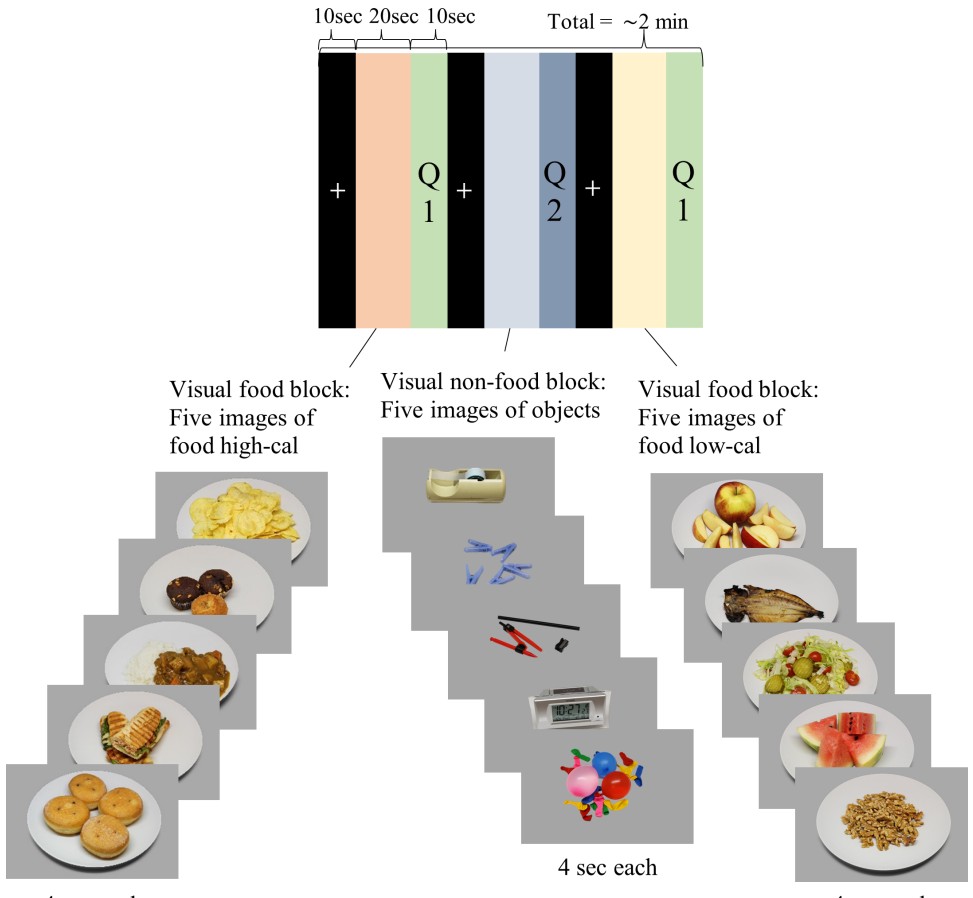

**Figure 2** Block design representing one of the six runs of the functional magnetic resonance imaging paradigm. Each run comprises three visual blocks and three question blocks. The visual blocks are 20s each and are composed of five pictures that change every 4s. The first block switches between high- and low-calorie foods, while the block of non-food objects is interleaved in between. Before each visual block takes place a fixation cross (10s) and after a question block (10s), with a question about the visual block that took place before. Food visual blocks will be followed by a decision-making question of whether the person would eat such foods, considering both the pleasure and health implications. On the other hand, non-food visual blocks will be followed by an attention question of whether the person recognises such objects. Cal, calorie; min, minutes; sec, seconds.

of health concerns. This information will also be used as a variable in the correlation analysis with central dopamine receptor function.

As was mentioned before, fMRI data using visual stimuli will be used to obtain a mean activation cluster related to visual food cue processing to use as mask in the quantitative analysis of PET brain images. Preprocessing of fMRI data includes head motion correction, mean intensity adjustment, cubic spine slice scan time correction, trilinear 3D motion correction and temporal filtering (high pass). Runs exceeding 3mm of movement in any axis will be excluded from further analysis. After inhomogeneity correction of structural MRI data, functional and structural MRI data will be coregistered and transformed to the standard Montreal Neurological Institute (MNI) space. A random effects General Linear Model (GLM) analysis will be computed for each subject to assess individual brain activity patterns. After that, to obtain activation clusters related to visual food cue processing, we will use contrasts between (high-calorie), (low-calorie) and (non-food objects). Finally, we will aggregate the

individual activation clusters in a mean activation cluster across groups to use as a mask in the PET analysis. Due to the low signal-to-noise ratio of [$^{11}$C]raclopride in extrastriatal brain regions,[24] we plan to be cautious while not ignoring that D2R are present with meaningful biological function outside the striatum.[25] All MRI data analysis will be performed using BrainVoyager 22.0.0 (Brain Innovation, Maastricht, The Netherlands).

### Setting for imaging acquisition (PET and MRI)

PET and MRI acquisition will be performed on the same day by dedicated and experienced technicians in the facilities of ICNAS. Due to the well-documented impact of the fasted versus fed state on responses to food cues, all subjects will perform MRI acquisition at the same time of the day, around 13:30 in a 6-hour (at least) fast state. To do so, participants will be instructed to have their last meal around 7:30. After the MRI scan, participants will consume the same standardised meal consisting of a vegetable soup, a sandwich and fruit to ensure a similar recent dietary intake for all participants. For PET acquisition,

participants will be instructed to abstain from alcohol for 24 hours, as well as from nicotine and caffeine for 12 hours before this acquisition.

## Statistical analyses

As already mentioned, mean regional binding values will be extracted from the D2/3R $BP_{ND}$ parametric maps of 23 individuals who have successfully responded to BS and 23 normal-weight healthy control individuals. Two one-sided test approaches will be performed to investigate if the binding of these receptors is the same in these two groups. To deal with multiple comparison issues, we will limit our hypotheses to three anatomical subregions of the striatum (caudate, putamen and ventral striatum) and a mean activation cluster across groups.

Since correlation is usually tested for two variables at a time, the association between the mean regional D2/3R $BP_{ND}$ values with anthropometric, neurobehavioural and clinical parameters will lead us to the same multiple comparison issues. Since we aim to test several variables, we will perform a correlation matrix where the mean regional D2/3R $BP_{ND}$ values of the three ROIs and the mean activation cluster across groups will be correlated with the age, hormonal status, five anthropometric metrics, neuropsychological scores and fMRI decision-making question responses. The generated matrix will allow us to see which pairs correlate most.

Finally, the exploratory analysis will compare independent groups of individuals to verify if obesity and the type of BS (sleeve gastrectomy and Roux-en-Y gastric bypass) display distinct mean regional D2/3R $BP_{ND}$ values. Since we are comparing independent groups of individuals, we will use multiple unpaired t-tests (for equal variance) or Welch's t-test (for unequal variance) followed by the Holm-Sidak method (to deal with multiple comparison issues). On the other hand, to track central dopaminergic changes resulting from BS, we will compare the same individuals before and after BS using analysis of variance: repeated measures, within–between interaction (for equal variance) or mixed-effects models (for unequal variance). We will perform Dunn's post hoc test to deal with multiple comparison issues in this case.

## Sample size justification

The sample size of 23 per group was estimated using the equation (5) published in Lakens[35] as the required to demonstrate the equivalence between two independent group means for an allocation ratio of 1, equivalence bound of 0.8236, α level of 0.05 and power of 80%.[35] The equivalence bound of 0.8236 was calculated by multiplying 1.16 (Cohen's d) with 0.71 (pooled SD). These values were set based on a meta-analysis performed to investigate group differences in case–control studies comparing the same central dopamine receptors between individuals with obesity and non-obese controls.[36] In the case of significant differences, the overall Cohen's d was $-1.16 \pm 0.71$.

## ETHICS AND DISSEMINATION

The project has received ethical approval from the local medical ethics committee of the Faculty of Medicine of the University of Coimbra (approval ID, CE_Proc. CE-088/202) and CHUC (approval ID OBS.SF.143/2021). Participants will sign an informed consent approved by these ethics committees. Results will be submitted for publication in international peer-reviewed journals and presented at relevant conferences. Summaries of the results will be provided to the participants.

**Contributors** MLP, JC and MC-B designed the protocol of the study. JC and MC-B designed the protocol of the MRI experiment. MLP and JC initiated the acquisition of data. MC-B supervised the project and provided the funding for PET and MRI acquisition and AA provided the expertise for [$^{11}$C]raclopride synthesis. MLP wrote the first draft of the manuscript. MLP, JC, AA and MC-B critically revised the manuscript and approved the final version.

**Funding** This Study Research Proposal was funded in 2020 by Fundação para a Ciência e a Tecnologia (FCT) for 4 years PhD Research Scholarship. Institutional funding is secured by Coimbra Institute for Biomedical Imaging and Translational Research (CIBIT) Funding (FCT/UIDP&B/4950/2020), CENTRO-01-0145-FEDER-000016, DSAIPA/DS/0041/2020 and ICNAS/ICNAS Pharma.

**Competing interests** None declared.

**Patient and public involvement** Patients were not involved in the development of the research question, outcome measures or study design but were involved in the dissemination.

**Patient consent for publication** Not applicable.

**Provenance and peer review** Not commissioned; externally peer reviewed.

**ORCID iD**
Miguel Castelo-Branco http://orcid.org/0000-0003-4364-6373

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
