## [Reviewer comments · BMJ Open]

ARTICLE DETAILS

TITLE (PROVISIONAL)	Portuguese observational cross-sectional clinical imaging study protocol to investigate central dopaminergic mechanisms of successful weight loss through bariatric surgery
AUTHORS	Pais, Marta Lapo; Crisóstomo, Joana; Abrunhosa, Antero; Castelo-Branco, Miguel

VERSION 1 – REVIEW

REVIEWER	Steven Fordahl UNC
REVIEW RETURNED	01-Nov-2023

GENERAL COMMENTS	Overall, the protocol is clear and straightforward. My only questions relate to how the MRI data are collected. Will there be any functional tests performed in the magnet to initiate dopaminergic brain regions? For example, images of food, choice tasks, or other reward related tasks, or will the MRI data be resting state baseline data only? Additionally, statistical analyses used for PET and fMRI data sets may need to be examined by modeling experts. Typically, fMRI studies use principal component analyses to determine how multiple variables inform BOLD activation. These statistical models are a little outside of my expertise, but reproducibility of the protocol could benefit from greater detail regarding the analyses.
--

REVIEWER	Valerie Darcey National Institutes of Health
REVIEW RETURNED	22-Nov-2023

GENERAL COMMENTS	Thank you for the opportunity to review this exciting observational protocol proposing to examine the molecular and functional correlates of successful weight loss from bariatric surgery. Applaud effort to publish protocols a priori for transparency and rigor! Such a study with an adequate sample size is needed for the field, however there are a number of issues related to clarity of information presented and some methodological concerns which dampen this reviewer's enthusiasm for now. Authors state that "major research question" is why some are successful with BS while others fail to respond yet their protocol only examines successful BS responders and is thus not designed to answer this particular research question. Page 4/13, line 36: Suggest revision of research question given study design: What the neurochemical correlates are of successful WL via BS is an outstanding research question. Again, in summary (p 10/13, line 29), this proposed work is not designed to "clarify the role of dopaminergic D2/3R signaling in BS resistance" (but rather to clarify
---

	the relationship of dopaminergic D2/3R availability and successful weight loss through BS. Primary hypothesis: DA neurotransmission is unaltered in successful BS [page 2/13, line 50]. As stated, this lacks clarity unfortunately, with respect to comparison group, definition of "success" and region+measure of interest. Suggest revising more like statement in p 8/13 line 8-9, "we expect the same BPnd values in individuals who successfully responded to BS and healthy controls" Secondary outcomes: DA status (defined how, and when... at baseline?) correlates with:  1. excess WL 2. total WL, 3. weight regain. a. These outcomes will be measured at an unstandardized timepoint post-op – time since surgery should be included as a covariate in analyses. Also what amount of weight regain is permitted while still meeting definition of "successful" BS to be included in patient group?  4. Neurobehavioral parameters 5. clinical parameters Authors have many (loosely defined) secondary outcomes pose an issue for multiple comparisons correction. Consider making some exploratory, particularly if not strong a priori hypotheses). Exploratory:  1. Whether [successful WL from] BS types (sleeve vs RYGB) display distinct DA signatures (define "DA signature" and when you will measure this? Variable post-op time points?) 2. if they [what exactly are DA signatures??] can be reversed after WL a. Again this is mentioned p 9/13, line 36 - what exactly does this mean, "if altered dopaminergic neurotransmission can be reversed after WL"? Authors primary outcome hypothesizes that there will be no difference in D2 BPND between successful BS post op and controls - are there pre-op raclopride PET scans already collected for these successful WL patients that authors are able to compare pre to post op to see if "altered" D2 BPND is "reversed" post-op? Suggest revising/clarifying language here. Page 3/13, line 12: As it takes years to typically develop obesity/comorbidities qualifying for BS, authors should recruit controls to match demographics of patients to minimize confound of age decreasing d2r availability. Reviewer notes this is included later in methods/participants, and suggests including this language earlier for reader's awareness. Page 3/13, line 14-15: Are these anatomical/ fMRI regions determined from the literature - or from an fMRI food based task localizer as indicated in page 3/13 line 43? Please clarify specific regions. Also, the sample size calculation is referring to the anticipated effect size between the means of ... which groups...and... for what measure (d2BP in what ROI)? Pre to post BS? Post BS vs normal weight control (which is specifically predicted to be "unaltered in successful BS", which presumably means authors expect no difference in D2BP between lean and successful BS?) I believe the manuscript/protocol would benefit tremendously from clarification on all these points.
--	---

Page 3/13, line 22-23 & page 3/13 line 48. Unfortunately, the way this interesting protocol is designed, authors will not be able to "identify patients who will benefit from the procedure" as they do not plan to include a group of BS patients that did not have "successful" weight loss. Suggest revising this language to better reflect investigation of BS/WL-associated changes in DA neurotransmission (not who is successful and who is not, since only "successful" patients included in protocol).

Page 3/13, line 37: This exploratory endpoint is not specifically included in page 2/13, line 56, unless this refers to 'whether [distinct DA signatures] can be reversed after weight loss'. Can keep language consistent perhaps?

Page 3/13 line 44. These food localizer-based ROIs will increase the statistical power by defining relevant regions at the *group level*, correct? Presumably individual food-localizer maps will be used to create a group mean ROI for group level analyses? Also (p 8/13, line 7), (unlike 18F-fallypride), 11C-Raclopride is generally considered not suitable to estimate extra-striatal BPND so suggest limiting PET ROIs to dorsal and ventral striatum. (Also, this is mentioned in section page 9/13 line 14+).

Page 5/13 line 8. Authors cite a landmark study (Wang 2001) but should be aware of new hypotheses and work in this area of BMI and D2R availability measured via PET (Janssen & Horstmann 2022; Darcey et al 2023;)

Page 5/13 line 43+: Comparison of R-en-Y and Sleeve listed as a secondary question here but indicated (correctly) as an exploratory comparison above. In fact, only correlations listed in this paragraph are listed as secondary outcomes above. Reviewer may have misunderstood what was inferred to be the primary aim but isn't it to compare the D2BP of those with successful BS-WL to normal weight controls (cross-sectional)? Reviewer inferred a secondary aim would be comparison of D2BP pre to post op (longitudinal). Suggest moving additional subgroup analyses AND correlational analyses to exploratory since it's not clear this study as presently specified is powered to detect these differences.

Page 6/13, line 5 & page 10/13 line 33: Mention of overweight but this group will not be included in study (inclusion criteria p 6/13 line 60).

Page 6/13 line 17+. This reviewer is still not clear on how authors intend on obtaining the pre-op imaging - is this a prospective study? How will you ensure that you are imaging BS patients who will be successful with WL? Perhaps an inclusion of a "Recruitment" section to describe how you will enroll prospective patients, collect pre-op scans, and analyze only successful WL?? But if this is the case, you may have an exploratory sized sample to generate hypotheses about D2 BPND in successful vs unsuccessful BS-WL.

Page 8/13 line 12+: If inclusion criteria (page 6/13, line 60) includes individuals only up to BMI 50, how will authors evaluate relationship between BPND and BMI in individuals with BMI>50? Also please refer to new hypotheses and work in this area of BMI and D2R availability measured via PET (Janssen & Horstmann 2022; Darcey et al 2023;) to further inform this hypothesis.

	Page 8/13, line 46+: Is this fMRI task previously published on? Can please include reference if so. If not, section would benefit from expansion on "brief" description of task. Event related or block design? How many food stimuli, duration of presentation? How will participants indicate "reported decision of whether eating or not such foods" (clarify language)? Is this a binary decision (yes/no) or a score? Will only images "selected [for eating]" be included in the food localizer because presumably images not selected [for eating] are not "eating-related"? Page 9/13 lines 13+: To minimize issues with multiple comparisons correction with an (as of yet) undetermined number of ROIs (including presumably some even outside the striatum), suggest limiting primary analysis to comparison of only D2 BPND in the 1. Dorsal Striatum and 2. Ventral striatum in 21 successful BS vs 21 age-gender matched controls in the same post-prandial state. Page 10/13, line5-6 & line 24: On its own C-11 raclopride will not be able to provide "brain maps of central dopamine receptors" but rather (critically) the map of D2/3R *availability* (given that BP is a function of both receptor density and competition between the radioligand with endogenous DA at the receptor). General methods comments:  1. Study will enroll only female participants but no plan to track menses indicated (or indication of effort to scan participants in same phase of menstrual cycle) given that DA neurotransmission fluctuates across cycle. 2. Recent dietary intake (even just fed vs fasted) can influence BPND. Suggest authors ensure participants are in a consistent postprandial state prior to PET scanning. 3. Definition of "successful" BS weight loss. Authors state > 50% EWL or 20% TWL between 1-5 years post-op, but what if a patient they enroll because of "success" at 1 year post op then goes on to regain weight making them ultimately unsuccessful? Suggest standardizing post-op timepoints to minimize this issue of definition. 4. More detail on imaging methods would be appreciated – what time of day will patients be scanned for PET and MR? Also authors mention subject specific neuroanatomy but no mention of parameters for T1 weighted (MPRAGE) high resolution image collection? Overall: Protocol manuscript would benefit from consistency and specificity of language across primary and secondary and exploratory aims (e.g. ClinicalTrials.gov definitions)
--	--

VERSION 1 – AUTHOR RESPONSE

Reviewer: 1

Dr. Steven Fordahl, UNC

Comments to the Author:

Overall, the protocol is clear and straightforward. My only questions relate to how the MRI data are collected. Will there be any functional tests performed in the magnet to initiate dopaminergic brain regions? For example, images of food, choice tasks, or other reward related tasks, or will the MRI data be resting state baseline data only? Additionally, statistical analyses used for PET and fMRI data sets may need to be examined by modeling experts. Typically, fMRI studies use principal component analyses to determine how multiple variables inform BOLD activation. These statistical models are a

little outside of my expertise, but reproducibility of the protocol could benefit from greater detail regarding the analyses.

RESPONSE: Thank you for pointing this out. Regarding MRI data collection, structural and fMRI data will be acquired on a 3T imaging system (MAGNETOM Prisma, Siemens Medical Solutions) using a 64-channel head-coil. First, we will acquire the structural MRI data using 3D-T1 weighted. After that, we will start the block-design fMRI paradigm (6 runs of 2 minutes each) using 2D-T2* weighted sequences. Each run comprises three visual blocks and three question blocks. The visual blocks have a duration of 20 seconds each and are composed of 5 pictures that change every 4 seconds. The first block is pictures of high or low caloric foods, the second is non-food objects and the third is the category of foods that did not appear in the first block. Before each visual block takes place a fixation cross (10 seconds) and after a question block (10 seconds), with a question about the visual block that took place before. Food visual blocks will be followed by a decision-making question of whether the person would eat such foods, considering both the pleasure aspect and the health implications. On the other hand, non-food visual blocks will be followed by an attention question of whether the person recognizes such objects. In the revised version, we rewrite the text to add all this information in the first paragraph of 'MRI Measurement (acquisition and analysis)' subsection of 'METHODS AND ANALYSIS'. We also included Figure 2 representing block-design one of the six runs of the fMRI paradigm.

We recognise that statistical analyses used for PET and fMRI needed some detailed technical information. In the revised manuscript, we clarify that the main outcome measure used in the statistical analyses - mean regional D2/3R BP_{ND} values - will be obtained through PET images and structural and functional MRI will be used in the analysis of PET images. In detail, structural MRI will be used for co-registration between PET and MRI images and fMRI to obtain functional brain regions related to eating behaviour. Since PET has limited spatial resolution, the co-registration is an important step to define anatomical brain regions in MRI images and use them as masks to extract the mean regional BP_{ND} values of those regions from PET images. On the other hand, block-design fMRI paradigm will be used to obtain subject specific functional activation clusters. We apologise that this part of the analysis lacked some detail in the last version of the manuscript. In the revised version we added relevant information regarding the analysis of block-design fMRI paradigm in the second paragraph of 'MRI Measurement (acquisition and analysis)' subsection as follows: "As was mentioned before, fMRI data using visual stimuli will be used to obtain eating-related subject-specific activation clusters to use as masks in the quantitative analysis of PET brain images. Pre-processing of fMRI data includes head motion correction, mean intensity adjustment, cubic spine slice scan time correction, trilinear 3D motion correction and temporal filtering (high pass). Runs exceeding 3 mm of movement in any axis will be excluded from further analysis. After inhomogeneity correction of structural MRI data, functional and structural MRI data will be coregistered and transformed to the standard MNI space. Thereafter, a random effects General Linear Model analysis will be computed, for each subject, to assess the brain activity patterns. Finally, to obtain the eating-related subject-

specific activation clusters we will use the conjunction contrast: [High & Low caloric foods > Objects] and Bonferroni correction to counteract the multiple comparisons with a fixed p-value of 0.05.”

The ‘Statistical analyses’ subsection of ‘METHODS AND ANALYSIS’ was also updated with detailed information about the statistical tests we will use and how we plan to deal with the issue of multiple comparisons.

Reviewer: 2

Dr. Valerie Darcey, National Institutes of Health

Comments to the Author:

Thank you for the opportunity to review this exciting observational protocol proposing to examine the molecular and functional correlates of successful weight loss from bariatric surgery. Applaud effort to publish protocols a priori for transparency and rigor! Such a study with an adequate sample size is needed for the field, however there are a number of issues related to clarity of information presented and some methodological concerns which dampen this reviewer’s enthusiasm for now.

Authors state that "major research question" is why some are successful with BS while others fail to respond yet their protocol only examines successful BS responders and is thus not designed to answer this particular research question. Page 4/13, line 36: Suggest revision of research question given study design: What the neurochemical correlates are of successful WL via BS is an outstanding research question. Again, in summary (p 10/13, line 29), this proposed work is not designed to "clarify the role of dopaminergic D2/3R signaling in BS resistance" (but rather to clarify the relationship of dopaminergic D2/3R availability and successful weight loss through BS).

RESPONSE: Thank you for your comment on this key aspect of the article. In the revised version of the manuscript, we clarify that our **main research question** is how brain dopamine receptors function in the context of successful WL through BS. Namely, determine if the binding of these receptors is similar in individuals who responded successfully to BS and age- and gender-matched normal-weight healthy individuals (first paragraph of ‘Study aims’ subsection). Regarding **secondary questions**, we aim to determine if the binding values of these receptors correlate with anthropometric metrics and other key variables such as neurobehavioral and clinical parameters (second paragraph of ‘Study aims’ subsection). Finally, **exploratory questions** will investigate whether the type of BS and obesity display distinct dopamine receptors binding values and track BS-associated central dopaminergic neurotransmission changes (third paragraph of ‘Study aims’ subsection). To make it easier to visualize the strategy we also improve Figure 1. In the revised version Figure 1 specifies the main and secondary study designs, research questions and samples used. This information was added and rewritten across the main text of the revised manuscript but it is particularly detailed in ‘Study aims’ subsection of ‘INTRODUCTION’, where the first, second and third paragraphs specify main, secondary and exploratory questions, respectively.

Primary hypothesis: DA neurotransmission is unaltered in successful BS [page 2/13, line 50]. As stated, this lacks clarity unfortunately, with respect to comparison group, definition of "success" and region+measure of interest. Suggest revising more like statement in p 8/13 line 8-9, "we expect the same BPnd values in individuals who successfully responded to BS and healthy controls"

RESPONSE: Across the revised version of the manuscript, we clarified that our “main research question”/ main goal is to investigate central dopamine receptors profiles in the context of successful weight loss through bariatric surgery. Namely, determine if the binding of these receptors is similar in individuals who responded successfully to BS and controls in caudate, putamen and ventral striatum and fMRI identified ROIs related to eating behaviour. We apologise for the lack of clarity in the last version of the manuscript regarding the comparison group, the definition of "success" and the region measure of interest. In the ‘Study design’ subsection of ‘METHODS AND ANALYSIS’ of the revised manuscript, we detailed the comparison group: “age- and gender-matched normal-weight (BMI: 18.5-24.9 Kg/m²) healthy individuals.”; and “success”: “To be considered successful, individuals should achieve at least 50% EWL% or 20% of WL% at 1 year post-op. [18] Since we will include participants from 1-5 years post-op, to verify this criterion we will use a retrospective approach and consider the weight participants had 1 year post-op. Additionally, we will also guarantee that at the time of data acquisition, participants still fulfil the criterion defined for 1 year after surgery (> 50% EWL% or 20% TWL%). Follow-up time post-op will be further used as a covariate in the statistical analysis.” The region measure of interest was clarified across the manuscript but particularly in the third paragraph of “PET Measurement (acquisition and analysis)” subsection: “Mean BPND values of anatomical brain ROIs will be extracted from the parametric maps. Due to the low signal-to-noise ratio of [11C]Raclopride in extrastriatal brain regions [23], anatomical ROIs will be defined in three subregions of the striatum, caudate, putamen and ventral striatum. Additionally, to increase the statistical power of the study analysis, we will also test subject-specific activation clusters that will result from fMRI analysis.”

Secondary outcomes: DA status (defined how, and when... at baseline?) correlates with:

1. excess WL
2. total WL,
3. weight regain.
 - a. These outcomes will be measured at an unstandardized timepoint post-op – time since surgery should be included as a covariate in analyses. Also what amount of weight regain is permitted while still meeting definition of "successful" BS to be included in patient group?
4. Neurobehavioral parameters
5. clinical parameters

Authors have many (loosely defined) secondary outcomes pose an issue for multiple comparisons correction. Consider making some exploratory, particularly if not strong a priori hypotheses).

RESPONSE: We removed the nomination of “DA status” from all text of the manuscript to avoid mixed interpretations. In the revised version is clarified that the main outcome measure used in the statistical analyses will be the mean striatal (caudate, putamen and ventral striatum) and functional regional D2/3R BPND values. In terms of timeline, this primary outcome measure will be acquired **just once** in individuals who have successfully responded to and controls and **twice** in individuals with obesity (before and after BS). Since we will include participants from 1-5 years post-op, follow-up time post-

op will be further used as a covariate in the statistical analysis. In this protocol study design the amount of weight regain is not used to define “success”. What we guarantee is that at the time of data acquisition, participants still fulfil the criterion defined for 1 year after surgery (> 50% EWL% or 20% TWL%), regardless of the amount of weight regained. Since neurobehavioral and clinical parameters lacked detail and transparency in the last version, we have rewritten the ‘Clinical and Neuropsychological Assessment’ subsection of ‘METHODS AND ANALYSIS’ to explicit which and how many variables we will include in this part.

Exploratory:

1. Whether [successful WL from] BS types (sleeve vs RYGB) display distinct DA signatures (define "DA signature" and when you will measure this? Variable post-op time points?)
2. if they [what exactly are DA signatures??] can be reversed after WL
 - a. Again this is mentioned p 9/13, line 36 - what exactly does this mean, "if altered dopaminergic neurotransmission can be reversed after WL"? Authors primary outcome hypothesizes that there will be no difference in D2 BPND between successful BS post op and controls - are there pre-op raclopride PET scans already collected for these successful WL patients that authors are able to compare pre to post op to see if "altered" D2 BPND is "reversed" post-op? Suggest revising/clarifying language here.

RESPONSE:

1. We removed the nomination of “DA signature” from all text of the manuscript to avoid mixed interpretations. In the revised version is clarified that the main outcome measure used in the statistical analyses will be the mean striatal and functional regional D2/3R BPND values. In the group of individuals who have successfully responded to BS this primary outcome measure will be defined in 1-5 years after BS. As such, follow-up time post-op will be further used as a covariate in the statistical analysis.

2. Thank you very much for your suggestion. Instead of writing that we will investigate whether binding values can be reversed after WL, we change to track central dopaminergic changes resulting from BS. To do so, we will include a cohort of individuals with obesity at two different time points, before and after BS. We believe that the binding of central dopamine receptors is altered in obesity and that these dysfunctional values can be reversed towards control condition values after WL. In the revised manuscript we clarify this issue.

Page 3/13, line 12: As it takes years to typically develop obesity/comorbidities qualifying for BS, authors should recruit controls to match demographics of patients to minimize confound of age decreasing d2r availability. Reviewer notes this is included later in methods/participants, and suggests including this language earlier for reader's awareness.

RESPONSE: Thank you for pointing this out. In the revised manuscript it is possible to find mention of controls in the abstract, study aims (INTRODUCTION), Study design and Statistical analyses (METHODS AND ANALYSIS).

Page 3/13, line 14-15: Are these anatomical/ fMRI regions determined from the literature - or from an fMRI food based task localizer as indicated in page 3/13 line 43? Please clarify specific regions. Also, the sample size calculation is referring to the anticipated effect size between the means of ... which groups...and... for what measure (d2BP in what ROI)? Pre to post BS? Post BS vs normal weight control (which is specifically predicted to be "unaltered in successful BS", which presumably means authors expect no difference in D2BP between lean and successful BS?) I believe the manuscript/protocol would benefit tremendously from clarification on all these points.

RESPONSE: The region measure of interest was clarified in the third paragraph of "PET Measurement (acquisition and analysis)" subsection as follows: "Mean BP_{ND} values of anatomical brain ROIs will be extracted from the parametric maps. Due to the low signal-to-noise ratio of [11C]Raclopride in extrastriatal brain regions [23], anatomical ROIs will be defined in three subregions of the striatum, caudate, putamen and ventral striatum. Additionally, to increase the statistical power of the study analysis, we will also test subject-specific activation clusters that will result from fMRI analysis."

To clarify how the sample size was calculated we have rewritten 'Sample size justification' subsection as follows: "The sample size of 23 per group was estimated using the equation (5) published in Lakens (2017) as the required to demonstrate the equivalence between two independent group means for an allocation ratio of 1, equivalence bound of 0.8236, α level of 0.05 and power of 80%. [30] The equivalence bound of 0.8236 was calculated by multiplying 1.16 (Cohen's d) with 0.75 (pooled standard deviation). These values were set based on a meta-analysis performed to determine group differences in case–control studies comparing the same central dopamine receptors between individuals with obesity and non-obese controls. [31] In the case of significant differences, the overall Cohen's d was -1.16 ± 0.71 ."

Page 3/13, line 22-23 & page 3/13 line 48. Unfortunately, the way this interesting protocol is designed, authors will not be able to "identify patients who will benefit from the procedure" as they do not plan to include a group of BS patients that did not have "successful" weight loss. Suggest revising this language to better reflect investigation of BS/WL-associated changes in DA neurotransmission (not who is successful and who is not, since only "successful" patients included in protocol).

RESPONSE: Thank you for noting this inconsistency in the protocol. We changed the manuscript accordingly.

Page 3/13, line 37: This exploratory endpoint is not specifically included in page 2/13, line 56, unless this refers to 'whether [distinct DA signatures] can be reversed after weight loss'. Can keep language consistent perhaps?

RESPONSE: To keep the language consistent across the manuscript we have taken into account Reviewer 2 suggestion and changed the abstract (page 2/13, line 56 of last version) to: "Finally, for

exploratory goals, we will include a cohort of individuals with obesity at two different time points, before and after BS. We aim to explore whether obesity and the type of BS (sleeve gastrectomy and Roux-en-Y gastric bypass) display distinct binding values for these receptors. Furthermore, we also intend to track central dopaminergic changes resulting from BS.”

Page 3/13 line 44. These food localizer-based ROIs will increase the statistical power by defining relevant regions at the *group level*, correct? Presumably individual food-localizer maps will be used to create a group mean ROI for group level analyses? Also (p 8/13, line 7), (unlike 18F-fallypride), 11C-Raclopride is generally considered not suitable to estimate extra-striatal BPND so suggest limiting PET ROIs to dorsal and ventral striatum. (Also, this is mentioned in section page 9/13 line 14+).

RESPONSE: The block-design fMRI paradigm will be used to obtain individualized functional ROIs at the *individual level* (subject-specific activation clusters). We apologise that this part of the analysis lacked detailed information in the last version of the manuscript. In the revised version we rewritten ‘MRI Measurement (acquisition and analysis)’ subsection as follows: “As was mentioned before, fMRI data using visual stimuli will be used to obtain eating-related subject-specific activation clusters to use as masks in the quantitative analysis of PET brain images... a random effects General Linear Model analysis will be computed, for each subject, to assess the brain activity patterns. Finally, to obtain the eating-related subject-specific activation clusters we will use the conjunction contrast: [High & Low caloric foods > Objects] and Bonferroni correction to counteract the multiple comparisons with a fixed p-value of 0.05.”

The suggestion was added in the third paragraph of “PET Measurement (acquisition and analysis)” subsection: “Mean BP_{ND} values of anatomical brain ROIs will be extracted from the parametric maps. Due to the low signal-to-noise ratio of [11C]Raclopride in extrastriatal brain regions [23], anatomical ROIs will be defined in three subregions of the striatum, caudate, putamen and ventral striatum.”

Page 5/13 line 8. Authors cite a landmark study (Wang 2001) but should be aware of new hypotheses and work in this area of BMI and D2R availability measured via PET (Janssen & Horstmann 2022; Darcey et al 2023;)

RESPONSE: The references suggested were included in the ‘Background’ subsection of ‘INTRODUCTION’ as follows: “Dopamine is an important neurotransmitter with implications in addiction as well as weight loss, decreased food intake and a reduced motivational drive to eat. [12]–[15] Concerning cerebral control of food intake, the dopamine type 2 receptors (D2R) are particularly interesting because drugs that block D2R increase appetite and result in weight gain [16] and treatment with dopamine receptor agonists with a high affinity for D2R causes weight loss. [12] Moreover, a study found that dietary fat restriction significantly decreased D2R availability in striatal clusters of individuals with obesity. [17]”

[15] - Janssen & Horstmann 2022

[17] - Darcey et al 2023

Page 5/13 line 43+: Comparison of R-en-Y and Sleeve listed as a secondary question here but indicated (correctly) as an exploratory comparison above. In fact, only correlations listed in this paragraph are listed as secondary outcomes above. Reviewer may have misunderstood what was inferred to be the primary aim but isn't it to compare the D2BP of those with successful BS-WL to normal weight controls (cross-sectional)? Reviewer inferred a secondary aim would be comparison of D2BP pre to post op (longitudinal). Suggest moving additional subgroup analyses AND correlational analyses to to exploratory since it's not clear this study as presently specified is powered to detect these differences.

RESPONSE: We clarify our main research question, secondary questions and exploratory analysis along the manuscript but it is particularly detailed in 'Study aims' subsection of 'INTRODUCTION', where the first, second and third paragraphs specify main, secondary and exploratory questions, respectively. We also changed Figure 1. In the revised version Figure 1 specifies the main and secondary study designs, research questions and samples used.

Page 6/13, line 5 & page 10/13 line 33: Mention of overweight but this group will not be included in study (inclusion criteria p 6/13 line 60).

RESPONSE: We have removed this mention.

Page 6/13 line 17+. This reviewer is still not clear on how authors intend on obtaining the pre-op imaging - is this a prospective study? How will you ensure that you are imaging BS patients who will be successful with WL? Perhaps an inclusion of a "Recruitment" section to describe how you will enroll prospective patients, collect pre-op scans, and analyze only successful WL?? But if this is the case, you may have an exploratory sized sample to generate hypotheses about D2 BPND in successful vs unsuccessful BS-WL.

RESPONSE: We apologize for the lack of clarity on this matter. In the revised version we elaborate on this explanation. We will include a cohort of individuals with obesity at two different time points, before and after BS (the longitudinal part of the Secondary study designs, Figure 1 was also updated accordingly). We don't have any way of knowing if they will be successful or not but we will just include at T2 – follow-up individuals who have successfully responded to BS. Being aware of this limitation we include in the 'Strengths and limitations of this study' the following: "Exploratory subgroup analysis is limited due to the dataset size, and in the longitudinal part of this analysis, some level of participant dropout or exclusion is to be expected"

Page 8/13 line 12+: If inclusion criteria (page 6/13, line 60) includes individuals only up to BMI 50, how will authors evaluate relationship between BPND and BMI in individuals with BMI>50? Also please refer to new hypotheses and work in this area of BMI and D2R availability measured via PET (Janssen & Horstmann 2022; Darcey et al 2023;) to further inform this hypothesis.

RESPONSE: We rewrite Page 8/13 line 12+ of the last version as follows: "...in exploratory analysis, we expect that obesity condition will be associated with altered mean BPND values in the brain but no differences between types of BS (sleeve gastrectomy and roux-en-Y gastric bypass). Moreover, we

expect that these dysfunctional values can be reversed to control condition values after WL." Since we do not intend to exclude this range of BMI a priori, we modified the BMI exclusion criteria of the obese group to BMI >35 kg/m².

Page 8/13, line 46+: Is this fMRI task previously published on? Can please include reference if so. If not, section would benefit from expansion on "brief" description of task. Event related or block design? How many food stimuli, duration of presentation? How will participants indicate "reported decision of whether eating or not such foods" (clarify language)? Is this a binary decision (yes/no) or a score? Will only images "selected [for eating]" be included in the food localizer because presumably images not selected [for eating] are not "eating-related"?

RESPONSE: We apologize for the lack of clarity on this matter. The fMRI task presented in this protocol is not published in any previous work of the institute. In the revised version we rewritten 'fMRI Measurement (acquisition and analysis)' subsection as follows: "...we will start the block-design fMRI paradigm (6 runs of ~2 minutes each) using 2D-T2* weighted sequences (72 interleaved slices; TE: 37ms; TR: 1000 ms; voxel size: 2x2x2 mm; FA: 68°; FOV: 200x200 mm). Each run comprised of three visual blocks and three question blocks begins with an instruction screen to test and set the directions (right and left) of the response buttons. The visual blocks have a duration of 20 seconds each and are composed of 5 pictures that change every 4 seconds. The first block is pictures of high or low caloric foods, the second is non-food objects and the third is the category of foods that did not appear in the first block. Before each visual block takes place a fixation cross (10 seconds) and after a question block (10 seconds), with a question about the visual block that took place before. Food visual blocks will be followed by a decision-making question of whether the person would eat such foods, considering both the pleasure aspect and the health implications. To answer the question the participant will use a response button to choose one of four options: "1- I certainly wouldn't eat it", "2- I probably wouldn't eat", "3- I would probably eat" and "I would certainly eat it". On the other hand, non-food visual blocks will be followed by an attention question of whether the person recognizes such objects. Here the options are the following: "1- I recognize all objects", "2- I recognize almost all objects"; "3- I hardly recognize any object" and "4- I don't recognize any objects". See Figure 2 for details." We also included Figure 2 representing block-design one of the six runs of the fMRI paradigm.

Page 9/13 lines 13+: To minimize issues with multiple comparisons correction with an (as of yet) undetermined number of ROIs (including presumably some even outside the striatum), suggest limiting primary analysis to comparison of only D2 BP_{ND} in the 1. Dorsal Striatum and 2. Ventral striatum in 21 successful BS vs 21 age-gender matched controls in the same post-prandial state.

RESPONSE: As suggested we have limited primary analysis to a comparison of only D2 BP_{ND} in the 1. Caudate, 2. Putamen and 3. Ventral striatum in 23 successful BS vs 23 age-gender matched controls.

Page 10/13, line5-6 & line 24: On its own C-11 raclopride will not be able to provide "brain maps of central dopamine receptors" but rather (critically) the map of D2/3R *availability* (given that BP is a function of both receptor density and competition between the radioligand with endogenous DA at the receptor).

RESPONSE: Instead of written "brain maps of central dopamine receptors" we clarify that [11C]Raclopride will be used to map brain dopamine type 2 and 3 receptors (D2/3R) non-displaceable binding potential (BP_{ND}).

General methods comments:

1. Study will enroll only female participants but no plan to track menses indicated (or indication of effort to scan participants in same phase of menstrual cycle) given that DA neurotransmission fluctuates across cycle.

RESPONSE: Thank you for pointing this out. In the revised version, we rewrite the text to add this information in 'Clinical and Neuropsychological Assessment' subsection of 'METHODS AND ANALYSIS' as follows: "As all our participants are female, and since dopaminergic neurotransmission fluctuates across age and the menstrual cycle, we will annotate the age and phase of the menstrual cycle for each participant. This information will be further used as a covariate in the statistical analysis."

2. Recent dietary intake (even just fed vs fasted) can influence BPND. Suggest authors ensure participants are in a consistent postprandial state prior to PET scanning.

RESPONSE: Eisenstein et al. (2020) when investigating striatal dopamine receptor binding using the same radiotracer found no differences during fasted and fed conditions. Yet, participants will be instructed to have their last meal around 7:30 and on the day of the visit participants will consume the same standardized meal consisting of a vegetable soup, a sandwich, and fruit to ensure a similar recent dietary intake for all participants. We have added a 'Setting for imaging acquisition (PET and MRI)' subsection with this information.

3. Definition of "successful" BS weight loss. Authors state > 50% EWL or 20% TWL between 1-5 years post-op, but what if a patient they enroll because of "success" at 1 year post op then goes on to regain weight making them ultimately unsuccessful? Suggest standardizing post-op timepoints to minimize this issue of definition.

RESPONSE: Thank you for this pertinent observation. We have rewritten that part of the text to clarify the definition of successful weight loss through bariatric surgery. In the revised manuscript, this change can be found in the text as follows: "as follows: "To be considered successful, individuals should achieve at least 50% EWL% or 20% of WL% at 1 year post-op. [18] Since we will include participants from 1-5 years post-op, to verify this criterion we will use a retrospective approach and consider the weight participants had 1 year post-op. Additionally, we will also guarantee that at the

time of data acquisition, participants still fulfil the criterion defined for 1 year after surgery (> 50% EWL% or 20% TWL%). Follow-up time post-op will be further used as a covariate in the statistical analysis.”

4. More detail on imaging methods would be appreciated – what time of day will patients be scanned for PET and MR? Also authors mention subject specific neuroanatomy but no mention of parameters for T1 weighted (MPRAGE) high resolution image collection?

RESPONSE: Thank you for pointing out that. We appreciate all your comments and suggestions to improve the manuscript. We have added a ‘Setting for imaging acquisition (PET and MRI)’ subsection with the following information: “PET and MRI acquisition will be performed in the same day by dedicated and experienced technicians in the facilities of ICNAS. Due to the well-documented impact of the fasted vs. fed state on responses to food cues, all subjects will perform MRI acquisition at the same time of the day, around 13:30 in a 6-hour (at least) fast state. To do so, participants will be instructed to have their last meal around 7:30. After MRI exam participants will consume the same standardized meal consisting of a vegetable soup, a sandwich, and fruit to ensure a similar recent dietary intake for all participants. For PET acquisition participants will be instructed to abstain from alcohol for 24 hours, as well as from nicotine and caffeine for 12 hours before this acquisition”.

Parameters for T1 weighted (MPRAGE) high-resolution image collection were also added: “...we will acquire the structural MRI data using 3D-T1 weighted sequence (192 slices; echo time (TE): 3.5 ms; repetition time (TR): 2530 ms; voxel size: 1x1x1 mm; flip angle (FA): 7°; field of view (FOV): 256x256 mm).”

VERSION 2 – REVIEW

REVIEWER	Valerie Darcey National Institutes of Health
REVIEW RETURNED	26-Jan-2024

GENERAL COMMENTS	Thank you for the opportunity to re-review this greatly improved protocol description! The authors satisfied many of the concerns of the previous submission with their revisions, additions, and clarifications. I look forward to seeing the outcome of this important work. Only a few new comments with this submission: 1. fMRI task a. Rather than labeling this as a task assessing eating behavior, I believe it would be a more accurate reflection of the task to label as ‘visual food cue processing’ or a ‘food localizer’ (locating where in the brain* participants are processing foods vs objects) Visual food cue processing is which is certainly related to food craving which is related to food intake/weight regulation, but as it is designed currently, there does not appear to be a “behavioral” component here (more on the “decision” question below). b. Pls indicate a few more details about the task to facilitate replicability: i. Whether the high kcal vs low kcal presentation order in the 6 fMRI
---

	runs be randomized and/or counterbalanced or will everyone receive alternating runs (high, low, high.. vs low, high, low...) ii. The images used by authors to select validated food images from previously published seem to be standardized which is applauded (given that low level visual quality differences may impact functional activation results from this planned contrast.) Can pls cite the source of these images? (e.g. Public repositories to examine their visual food cue processing). c. "...a decision making question of whether the person would eat such foods considering both the pleasure and health implications." i. This is still unclear to me. What exactly is the objective of the decision question after the blocks? 1. Is the point to obtain ratings to be used to parametrically convolve with the hemodynamic response function from the food block? a. How do you know whether/which foods participants are considering (or giving more weight to) when they answer this? what exactly is the objective of this question block? 2. If the objective is to get participants to *pay attention* to the food block maybe asking a question orthogonal to the design (e.g., did you see a [THIS FOOD] among the last set of images? Definitely yes, Definitely no, Unsure/maybe etc etc). That way could have an equivalent question to ensure attention in the Objects block. While not specifically validated for psychometric properties, you might be able to generate some hypotheses with the accuracy data (attentional bias to the high kcal vs low kcal foods) etc. d. Will subject specific activation clusters from the block design visual food cue task be aggregated over the group to generate a group mask of mean size/shape? i. Potential issues to extract individual binding potential cluster mean per individual to use in group analysis given that the size, shape, precision location (and therefore density of D2R) of each cluster will likely vary between subjects? ii. Also, given the imprecision with which raclopride can measure D2BP outside of high D2 dense regions (striatum), what will the authors do if this functional cluster falls outside of the striatal gray matter? iii. Perhaps authors would consider a mean functional ROI across individuals to use as an ROI for PET but caution against use outside of striatum for reason above. 2. Citations a. Page 3/31 line 57 - authors please check reference 16, this study is not a drug study? b. Additionally, reference 12 cites a commentary, not the primary research – I mention this in case this was in error? c. Page 3 line 59 - thank you for trying to cite my work on diet manipulation! However I suppose I was trying to draw your attention instead more towards latest finding in this preprint (still under review). You may be interested to learn that our data suggest that there may be more dopamine tone (not fewer receptors) at higher BMI. Curious how WL achieved through BS impacts this! i. https://www.medrxiv.org/content/10.1101/2023.09.27.23296169v1
--	---

VERSION 2 – AUTHOR RESPONSE

Reviewer: 2

Dr. Valerie Darcey, National Institutes of Health Comments to the Author:

Thank you for the opportunity to re-review this greatly improved protocol description! The authors satisfied many of the concerns of the previous submission with their revisions, additions, and

clarifications. I look forward to seeing the outcome of this important work. Only a few new comments with this submission:

1.fMRI task

a. Rather than labeling this as a task assessing eating behavior, I believe it would be a more accurate reflection of the task to label as 'visual food cue processing' or a 'food localizer' (locating where in the brain* participants are processing foods vs objects) Visual food cue processing is which is certainly related to food craving which is related to food intake/weight regulation, but as it is designed currently, there does not appear to be a "behavioral" component here (more on the "decision" question below).

RESPONSE: We agree. Accordingly, we have modified the nomination of "eating behaviour" to "visual food cue processing" in the fMRI task.

b.Pls indicate a few more details about the task to facilitate replicability:

i. Whether the high kcal vs low kcal presentation order in the 6 fMRI runs be randomized and/or counterbalanced or will everyone receive alternating runs (high, low, high.. vs low, high, low...)

RESPONSE: Thank you for your suggestion. In the revised version, we rewrite the text to add this information in 'MRI Measurement (acquisition and analysis)' subsection of

'METHODS AND ANALYSIS' as follows: "For all participants, the sequence of the fMRI task is the same, where the first block switches between high- and low-calorie foods, while the block of non-food objects is interleaved in the middle."

ii.The images used by authors to select validated food images from previously published seem to be standardized which is applauded (given that low level visual quality differences may impact functional activation results from this planned contrast.) Can pls cite the source of these images? (e.g. Public repositories to examine their visual food cue processing).

RESPONSE: Thank you for pointing this out. We added the mentioned reference in 'MRI Measurement (acquisition and analysis)' subsection of 'METHODS AND ANALYSIS' as follows: "In each visual block (20 seconds), participants will be presented with 5 images of low or high-calorie foods or non-food objects from the Cross Cultural Food Images Database (CROCUFID).[34]"

[34] - A. Toet et al., "CROCUFID: A Cross-Cultural Food Image Database for Research on Food Elicited Affective Responses," *Front Psychol*, vol. 10, no. JAN, Jan. 2019, doi: 10.3389/fpsyg.2019.00058.

c."...a decision making question of whether the person would eat such foods considering both the pleasure and health implications."

i.This is still unclear to me. What exactly is the objective of the decision question after the blocks?

1.Is the point to obtain ratings to be used to parametrically convolve with the hemodynamic response function from the food block?

RESPONSE: We plan to use the decision-making questions to evaluate how people rate rewards in terms of pleasure and health concerns. Namely, it is important to achieve information on dissociable psychological components of reward, particularly 'liking' (hedonic impact) and 'wanting' (incentive salience). This information will be combined with the neuropsychological scores to conduct a correlation analysis with central dopamine receptor function. Additionally, after the fMRI acquisition, the participants will use for rating a horizontal, 100-mm, bidirectional scale between the extremes "I don't like it at all" and "I like it a lot" to rate the food images in terms of preferences regardless of health concerns. This information will also be used as a variable in the correlation analysis with central dopamine receptor function. This information was added to the 'MRI Measurement (acquisition and analysis)' subsection of 'METHODS AND ANALYSIS'.

a. How do you know whether/which foods participants are considering (or giving more weight to) when they answer this? what exactly is the objective of this question block?

RESPONSE: Thank you for your question. As mentioned above, after the fMRI acquisition, the participants will use for rating a horizontal, 100-mm, bidirectional scale between the extremes "I don't like it at all" and "I like it a lot" to rate the food images in terms of preferences regardless of health concerns. This information will be used as a variable in the correlation analysis with central dopamine receptor function. This information was added to the 'MRI Measurement (acquisition and analysis)' subsection of 'METHODS AND ANALYSIS'.

2.If the objective is to get participants to *pay attention* to the food block maybe asking a question orthogonal to the design (e.g., did you see a [THIS FOOD] among the last set of images? Definitely yes, Definitely no, Unsure/maybe etc etc). That way could have an equivalent question to ensure attention in the Objects block. While not specifically validated for psychometric properties, you might be able to generate some hypotheses with the accuracy data (attentional bias to the high kcal vs low kcal foods) etc.

RESPONSE: As detailed above, decision-making questions were not designed to be an attention task. We now explain that we were more interested in the dissociable psychological components of reward: 'liking' (hedonic impact), 'wanting' (incentive salience), and learning (predictive associations and cognitions). In particular, "liking" and "wanting" were used in our design.

d.Will subject specific activation clusters from the block design visual food cue task be aggregated over the group to generate a group mask of mean size/shape?

i. Potential issues to extract individual binding potential cluster mean per individual to use in group analysis given that the size, shape, precision location (and therefore density of D2R) of each cluster will likely vary between subjects?

RESPONSE: Thank you for pointing this out. Following the reviewer's suggestion to use a mean functional ROI across individuals (question below iii) in the revised version of the manuscript, we clarify that we intend to generate a group mask of mean striatal activation clusters. This information was added to the 'MRI Measurement (acquisition and analysis)' subsection of 'METHODS AND ANALYSIS' as follows: "A random effects General Linear Model (GLM) analysis will be computed for each subject to assess individual brain

activity patterns. After that, to obtain activation clusters related to visual food cue processing, we will use contrasts between [High-calorie], [Low-calorie] and [Non-food objects]. Finally, we will aggregate the individual activation clusters in a mean activation cluster across groups to use as a mask in the PET analysis. Due to the low signal-to-noise ratio of [11C]raclopride in extrastriatal brain regions[24], we plan to be cautious while not ignoring that D2R are present with meaningful biological function outside the striatum.[25]"

ii. Also, given the imprecision with which raclopride can measure D2BP outside of high D2 dense regions (striatum), what will the authors do if this functional cluster falls outside of the striatal gray matter?

RESPONSE: As mentioned in the above response, we plan to be cautious while not ignoring that D2R are present with a meaningful biological function outside the striatum. We now provide prior references on this issue:

-[25]D. Rebelo, F. Oliveira, A. Abrunhosa, C. Januário, J. Lemos, and M. Castelo-Branco, "A link between synaptic plasticity and reorganization of brain activity in Parkinson's disease," Proceedings of the National Academy of Sciences, vol. 118, no. 3, Jan. 2021, doi: 10.1073/pnas.2013962118.

iii. Perhaps authors would consider a mean functional ROI across individuals to use as an ROI for PET but caution against use outside of striatum for reason above.

RESPONSE: Thank you for your suggestion. We added this suggestion: "Finally, we will aggregate the individual activation clusters in a mean activation cluster across each group to use as a mask in the PET analysis. Due to the low signal-to-noise ratio of [11C]raclopride in extrastriatal brain regions[24], we plan to be cautious while not ignoring that D2R are present with meaningful biological function outside the striatum.[25]"

2. Citations

a. Page 3/31 line 57 - authors please check reference 16, this study is not a drug study?

RESPONSE: Thank you for pointing this out. We modified the reference for an original study (see below).

[16]M.-J. Poulin et al., “Management of Antipsychotic-Induced Weight Gain: Prospective Naturalistic Study of the Effectiveness of a Supervised Exercise Programme,” Australian

& New Zealand Journal of Psychiatry, vol. 41, no. 12, pp. 980–989, Dec. 2007, doi: 10.1080/00048670701689428.

b. Additionally, reference 12 cites a commentary, not the primary research – I mention this in case this was in error?

RESPONSE: Thank you for your observation. We agree that it is better to cite primary research, so we changed the reference to the one below as it is more suitable.

- C. Figueira et al., “Hypothalamic dopamine signalling regulates brown fat thermogenesis,” Nat Metab, vol. 1, no. 8, pp. 811–829, Aug. 2019, doi: 10.1038/s42255-019-0099-7.

c. Page 3 line 59 - thank you for trying to cite my work on diet manipulation! However I suppose I was trying to draw your attention instead more towards latest finding in this preprint (still under review). You may be interested to learn that our data suggest that there may be more dopamine tone (not fewer receptors) at higher BMI. Curious how WL achieved through BS impacts this!

i. <https://www.medrxiv.org/content/10.1101/2023.09.27.23296169v1>

RESPONSE: Thank you for your reading suggestion. In the revised version, we rewrite the text to add this information in 'PET Measurement (acquisition and analysis)' subsection of 'METHODS AND ANALYSIS' as follows: "It is important to note that because [11C]raclopride is susceptible to endogenous dopamine, D2/3R BPND values may not reflect only receptor availability but also the levels of dopamine tone. A recent preprint study compared [11C]raclopride to [18F]fallypride, a radiotracer also susceptible to endogenous dopamine but with a higher affinity for D2/3R, and concluded that D2/3R BPND values measured by PET with [11C]raclopride are more sensitive for revealing between-group differences resulting from dopamine tone.[26]"

Reviewer: 2

Competing interests of Reviewer: None.

VERSION 3 – REVIEW

REVIEWER	Valerie Darcey National Institutes of Health
REVIEW RETURNED	27-Feb-2024
GENERAL COMMENTS	Appreciate authors addressing this reviewer’s comments on this important protocol! I’m of the opinion that there remains only one point of clarification. Based on their text, authors state that the *singular* question that appears after a block of *many* food images will be, “would you eat such foods, considering both pleasure and health implications”. They add, however, that their stated goal for asking this *singular* question after high calorie and low calorie

	image blocks is to “achieve information on *dissociable* psychological components of reward, particularly liking (hedonic impact) and wanting (incentive salience)”. I agree that liking and wanting are important components of reward to assess but I believe it will be difficult to disentangle these two constructs from a single question worded in this way. Do authors have knowledge of studies using a similarly worded question to assess both liking and wanting? I see that “liking” will be assessed outside of the scanner using VAS for each of the food images. Perhaps either the authors can add a parallel “wanting” assessment outside the scanner (i.e., wanting for each food image) or ask at least 2 questions after each block of images during fMRI acquisition (i.e., one question for each of 2 constructs of interest, liking and wanting, e.g., How much would you want to eat these foods right now? How much do you like these foods?).
--	---

VERSION 3 – AUTHOR RESPONSE

We want to thank Reviewer 2 for the relevant question regarding how we will deal with two psychological components of reward: liking (hedonic impact) and wanting (incentive salience). There was a typo in the text, misleading to the wrong idea. We do not intend to disassociate these two psychological rewards components but to integrate them. We have changed the manuscript accordingly as follows: “and in particular to achieve integrated information on psychological components of reward, particularly liking (hedonic impact) and wanting (incentive salience).”

We believe that this new version is suitable for publication. Please let us know if you still have any questions or concerns about the manuscript. We will be happy to address them.